# Effect of Yeast *Saccharomyces cerevisiae* as a Probiotic on Diet Digestibility, Fermentative Metabolites, and Composition and Functional Potential of the Fecal Microbiota of Dogs Submitted to an Abrupt Dietary Change

**DOI:** 10.3390/microorganisms11020506

**Published:** 2023-02-17

**Authors:** Taís Silvino Bastos, Camilla Mariane Menezes Souza, Héloïse Legendre, Nadège Richard, Rachel Pilla, Jan S. Suchodolski, Simone Gisele de Oliveira, Achraf Adib Lesaux, Ananda Portella Félix

**Affiliations:** 1Department of Animal Science, Federal University of Paraná, 1540, Curitiba 80035-050, PR, Brazil; 2Phileo by Lesaffre, 137 Rue Gabriel Péri, 59700 Marcq-en-Baroeul, France; 3Gastrointestinal Laboratory, Department of Small Animal Clinical Sciences, Texas A&M University, College Station, TX 77843-4474, USA

**Keywords:** biogenic amines, dysbiosis index, gastrointestinal functionality, metagenomics

## Abstract

The aim was to evaluate the effects of yeast probiotic on diet digestibility, fermentative metabolites, and fecal microbiota of dogs submitted to dietary change. Sixteen dogs were divided into two groups of eight dogs each: control, without, and with probiotic, receiving 0.12 g/dog/day of live *Saccharomyces cerevisiae* yeast. The dogs were fed a lower protein and fiber diet for 21 days and then changed to a higher protein and fiber diet until day 49. Yeast supplementation did not statistically influence diet digestibility. The probiotic group had a lower fecal concentration of total biogenic amines (days 21 and 49), ammonia (day 23), and aromatic compounds and a higher fecal concentration of butyrate (*p* < 0.05). The probiotic group showed a lower dysbiosis index, a higher abundance (*p* < 0.05) of *Bifidobacterium* (days 35 and 49) and *Turicibacter,* and a lower abundance of *Lactobacillus* and *E. coli* (*p* < 0.05). Beta diversity demonstrated a clear differentiation in the gut microbiota between the control and probiotic groups on day 49. The control group showed upregulation in genes related to virulence factors, antibiotic resistance, and osmotic stress. The results indicated that the live yeast evaluated can have beneficial effects on intestinal functionality of dogs.

## 1. Introduction

Although it may be common among pet owners, abrupt dietary changes for dogs can contribute to digestive disorders, gastrointestinal discomfort, and losing stool. In addition, it may also affect intestinal integrity, gut microbiota, and fermentation of end-products [1].

Functional additives have been studied in pet nutrition and have demonstrated their ability to mitigate gastrointestinal disorders triggered by dietary changes. Thus, probiotics based on *Saccharomyces cerevisiae* yeast may positively influence intestinal and microbial function, resulting in less digestive discomfort in such situations [1,2]. Although the mechanisms by which yeast probiotics may contribute to gastrointestinal functionality are not fully elucidated, it is possible that they act mainly through the production of metabolites, such as short-chain fatty acids (SCFA), antioxidants, and B-vitamins and directly interact with other microorganisms through its cell-wall-derived mannan-oligosaccharides (MOS) and β-glucans [1,3]. The dietary supplementation of live *Saccharomyces cerevisiae* reduced *Escherichia coli* counts in the feces of dogs [4] and modulated the immunity and the intestinal microbiota, improving microbial diversity in piglets [5,6,7].

However, despite these potential benefits, to the best of our knowledge, we did not find studies that evaluated the gut microbiota and its functional genes through advanced molecular tools in dogs receiving live yeast. In addition, the evaluation of relatively new outcomes, such as the dysbiosis index [8] may also help elucidate the effects of yeast probiotic supplementation in dogs. The present study aimed to evaluate the digestibility of nutrients and energy, intestinal fermentative products, and fecal microbiota and its functional genes through Kyoto Encyclopedia of Genes and Genomes (KEGG) orthology (KO) in dogs submitted to an abrupt dietary change with or without the supplementation of a live yeast probiotic.

## 2. Materials and Methods

### 2.1. Animals and Housing

Sixteen adult intact beagle dogs (5 years of age) were used (eight males and eight females), with an average body weight of 11.61 ± 0.12 kg, and body condition score of 6 ± 0.12 [1–9 scale] [9]. All animals underwent previous clinical examinations and were individually housed in covered kennels (5 m long × 2 m wide), containing a bed and free access to fresh water. The kennels had side wall grates allowing visual and limited interaction with neighboring dogs. The environment temperature ranged from 14 °C to 25 °C with a 12 h light-dark cycle (light 6 am–6 pm). Dogs had free access to an outdoor area under supervision for 2 h per day, except during fecal collection days. All dogs received extra attention during fecal collection periods. Throughout the study, all dogs remained under the supervision of research staff and of the veterinarian responsible for the laboratory.

### 2.2. Experimental Diets and Groups

The experiment evaluated two different diets with or without supplementation of a yeast probiotic (1 × 10^10^ CFU/g *Saccharomyces cerevisiae.* CNCM I-5660) product (Actisaf^®^, Phileo by Lesaffre, Marcq-en-Barœul, France) in two periods. The first diet was a commercial adult dog diet with lower protein (20.42%) and fiber (6.10%) concentrations (LPF). The second diet was a commercial diet for weight loss of adult dogs with higher protein (27.52%) and fiber (27.20%) concentrations (HPF). Both diets did not present functional additives such as zeolite, yeast products, prebiotics, probiotics, yucca, etc. (Table 1).

The LPF diet was offered during days 1 to 21 to the 16 dogs, divided into groups of 8 dogs each (4 males and 4 females in each group): control, without yeast supplementation and probiotic, with supplementation of 0.12 g yeast probiotic per dog per day. The yeast product was weighed daily on a precision scale (Mettler Toledo ME203 scale, São Paulo, Brazil) and individually applied on top of each diet. On day 22 of the experiment, dogs were abruptly transitioned to the HPF diet without adaptation. The two experimental groups (control and probiotic) continued to receive the yeast probiotic product (probiotic) or not (control) during days 22 to 49.

Both diets were offered in amounts to keep the dog’s weight adjusting to the recommendations for the maintenance of adult dogs [10]. Diets were offered twice a day (8:30 a.m. and 4:00 p.m.). Water was provided ad libitum.

Main ingredients LPF: Corn, poultry by-products meal, meat and bone meal, soybean meal, chicken hydrolysate, poultry fat, vitamins, and minerals. HPF: Corn, poultry by-products meal, corn gluten meal, soy hulls, cellulose, chicken and swine hydrolysate, poultry fat, soybean oil, linseed, vitamins, and minerals.

### 2.3. Diet Digestibility

On days 17 (after 8:30 am) to 22 (until 8:30 am), total feces were collected from each individual dog for evaluation of digestibility of the LPF diet. On day 22, at 8:30 am, dogs were abruptly transitioned to the HPF diet. From days 45 (after 8:30 am) to 50 (until 8:30 am), total feces were collected from each individual dog for evaluation of digestibility of the HPF diet. During both periods, feces were collected and weighed at least two times per day and stored in individual plastic containers in a freezer (−20 °C) for later analysis.

At the end of each collection period, feces of each replicate were thawed at room temperature and homogenized separately, forming a composite fecal sample from each animal. Feces were dried in a forced ventilation oven (320-SE, Fanem, São Paulo, Brazil) at 55 °C for 48–72 h or until reaching constant weight. The diets and feces were ground in a hammer mill (Arthur H. Thomas Co., Philadelphia, PA, USA) using a 1.0 mm wire mesh sieves for the bromatological testing.

The amounts of dry matter (DM) at 105 °C (DM105), crude protein (CP, method 954.01), ether extract in acid hydrolysis (EE, method 954.02), and ash (942.05) were determined in both diets and feces according to [11]. The total dietary fiber (TDF), insoluble fiber (IF), and soluble fiber (SF) of the diets were analyzed according to [12]. The amount of gross energy (GE) was established using a calorimetric pump (Parr Instrument Co., model 1261, Moline, IL, USA), and organic matter (OM) was calculated by the difference between 100—Ash.

### 2.4. Fecal Fermentative Products Analysis

On days 0, 21, 23, 35, and 49 one fresh fecal sample from each dog was collected within 15 min of defecation to analyze pH, score, DM, ammonia [13], phenols, indoles, biogenic ammines [14], SCFA, branched-chain fatty acids (BCFA), and microbiome and functional analysis. Fecal samples were scored according to a 5-point scale: 1 = feces are soft and have no defined shape; 2 = feces are soft and poorly formed; 3 = feces are soft, formed and moist; 4 = feces are well formed and consistent; 5 = feces are well formed, hard and dry [15]. Fecal pH was measured using a digital pH meter (331, Politeste Instrumentos de Teste Ltd.a, São Paulo, SP, Brazil) using 3.0 g of fresh feces diluted with 30 mL of distilled water. Fecal DM (DM55) was determined after drying in a forced ventilation oven (320-SE, Fanem, São Paulo, Brazil) at 55 °C for 48–72 h or until reaching constant weight. After dried, feces were ground in a hammer mill (Arthur H. Thomas Co., Philadelphia, PA, USA), using 1.0 mm sieve and analyzed for DM at 105 °C (DM105) [11]. Total fecal DM was calculated as: (DM55 × DM105)/100.

For SCFA and BCFA determination, 10 g of fecal sample was weighed and mixed with 30 mL of 16% formic acid. This mixture was homogenized and stored at 4 °C for a period of 3 to 5 days. Before the analysis, these solutions were centrifuged at 2500 *g* in a centrifuge (2K15, Sigma, Osterodeam Hans, Germany) for 15 min. At the end of the centrifugation, the supernatant was separated and subjected to further centrifugation at the same g and time. Each sample underwent three centrifugations, and the final supernatant was transferred to duly identified microtubes and frozen at −14 °C. Subsequently, the samples were thawed and centrifuged again at 18,000 g for 15 min. (Rotanta 460 Robotic, Hettich, Tuttlingen, Germany). Fecal SCFA and BCFA were analyzed by gas chromatography (Shimadzu, model GC-2014, Kyoto, Japan) by using a glass column (Agilent Technologies, HP INNO wax-19091N, Santa Clara, CA, USA) of 30 m long and 0.32 mm wide. Nitrogen was used as the carrier gas at a flow rate of 3.18 mL/min. Working temperatures were 200 °C in the injection, 240 °C in the column (at a rate of 20 °C/min), and 250 °C in the flame ionization detector.

Phenols and indoles were analyzed by chromatography, with a GCMS2010 Plus gas chromatographer (Shimadzu, Kyoto, Japan^®^), coupled to a TQ8040 mass spectrometer with an AC 5000 autosampler and a split–splitless injector. Chromatographic separations were obtained in the SH-Rtx-5MS (30 m × 0.25 mm × 0.25 μm—Shimadzu^®^) column with a 1.0 mL min^−1^ flow rate, and helium as the drag gas at a 5.0 rate. The transfer line and ionization source temperatures were maintained at 40 °C and 220 °C, respectively, the 1 L injection volume in the split mode (1:10 rate). The GC oven temperature was maintained at 220 °C (5 min), with a 40 °C min^−1^ increase to 280 °C (5 min). Total analysis time was 31 min, and the mass spectrometer operated in the full scan modes (m/z = 40 to 400) and selective ion monitoring (SIM), electron ionization at 70 eV. GCMS solution^®^ (version 4.30, Shimadzu, Kyoto, Japan) was the software used in the data analysis.

### 2.5. Microbiome and Functional Analysis

Fecal DNA was extracted with the PowerSoil Pro extraction kit (Qiagen, Venlo, The Netherlands). Fecal DNA was sequenced at Diversigen using BoosterShot Shallow Shotgun Sequencing. Briefly, for sequencing libraries preparation, the Nextera XT DNA Library Preparation Kit (Illumina Inc., San Diego, CA, USA) was used before the libraries were pooled. After this step, SPRI bead purification and concentration were processed using SpeedBeads Magnetic Carboxylate Modified Particles (Cytiva Life Sciences, Marlborough, MA, USA). The resulting pooled libraries were denatured by NaOH before being diluted and spiked by 2% PhiX.

The metagenomic sequencing was performed on an Illumina NovaSeq 6000 System using a single-end 1 × 100 base pair read chemistry), followed by being multiplexed on the sequencer before converting to FASTQ files and filtering for low quality (Q-score < 30) and length (<50). Adapter sequences were trimmed, and all sequences were trimmed to a maximum length of 100 bp before alignment. The raw sequences were made using NCBI Sequence Read Archive before analysis with established pipelines. In terms of taxonomic classification, FASTA sequences were aligned making a curated database, which contained all representative genomes in the NCBI RefSeq, a representative genome collection for prokaryotes with additional manually curated strains for bacteria [16]. Alignments were made at 97% identity and compared to reference genomes. Every input sequence was compared to every reference sequence in the Diversigen DivDB-Dog database using fully gapped alignment with BURST. Ties were broken by minimizing the overall number of unique Operational Taxonomic Units (OTUs). The input sequences were listed as the lowest common ancestor for taxonomy assignment, which was not compatible with <80% of the reference sequences. OTUs accounting for less than one million of all species-level markers and OTUs with <0.01% of their unique genome regions matching as well as <0.1% of the whole genome were discarded. The average sequencing depth was 2.3 ± 0.8 million (mean ± standard deviation) reads per sample.

For downstream analysis, normalized and filtered tables were used in QIIME2 [17]. The OTU tables were rarified at 158,023 sequences per sample, which was the lowest read of the samples. Alpha diversity was evaluated by the number of species, Shannon–Wiener index, and Chao1 index, and observed operational taxonomic units (OTUs) using the rarefied OTU table. Beta diversity was evaluated by principal coordinate analysis (PCoA) plots using the Bray–Curtis dissimilarity.

KO groups were observed with alignment at 97% identity against a gene database derived from the NCBI RefSeq representative genome collection for prokaryotes with additional manually curated strains for bacteria mentioned above for functionally annotated genes [18,19]. A rarefaction depth of 42,945 sequences per sample was used. The directly observed KO counts reported as relative abundance within each sample were expressed in a KO table and downstream tables. KOs were then collapsed to level-2 and level-3 KEGG pathways and KEGG modules.

The qPCR analysis of *Faecalibacterium*, *Blautia*, *Fusobacterium*, *Turicibacter*, *Clostridium hiranonis*, *Streptococcus*, *Escherichia coli*, and total bacteria for calculation of dysbiosis index was performed according to [8].

### 2.6. Calculations and Statistical Analysis

Based on the laboratory results, the coefficients of total tract apparent digestibility (ATTD) and the diet metabolizable energy (ME) were calculated according to [20]:ATTD (%) = [(g of nutrient intake − g of nutrient excretion)/g of nutrient intake] × 100. ME (kcal/kg) = {kcal/g GE intake − kcal/g GE fecal excretion − [(g CP intake − g CPfecal excretion) × 1.25kcal/g]}/g of feed intake.

Main statistical analyses were performed using Minitab^®^ (version 19.2020.1) software. Each dog was an experimental unit. Yeast supplementation and sex were treated as fixed effects; the day of measurement was treated as a repeated measure. Observations that were at least 1.5 times the interquartile range were considered as possible outliers. A Grubbs’ test was performed for confirmation if only one possible outlier was detected. If more than one possible outlier was detected, a Rosner test was performed using the function “rosnertest” from package “EnvStats” (Millard 2013) with R software (version 4.0.2). An analysis of variance (two-way ANOVA) was performed, using the function General Linear Model, to assess the effects of supplementation and interaction with the day of measurement. Normality and homoscedasticity assumptions were visually checked using residuals plots. Data were log-transformed if residuals were not uniformly distributed. Fisher LSD with Bonferroni correction was applied as post hoc test. Statistical significance was defined as *p* value < 0.05 while *p* < 0.10 was considered a trend.

Alpha diversity metrics for the microbiome results were evaluated by Chao1 and Shannon indexes using QIIME2 v.2021.2. Beta diversity, using Bray–Curtis distances, and its plots were analyzed and generated using Past software 4.03. Analysis of similarity (ANOSIM) was used to evaluate the similarity of the microbiota and KO terms between groups at different time points.

Differential abundance of KEGG Modules and KO terms between control and probiotic groups on days 21 and 49 and between diets (day 21 vs. day 49 regardless of the treatment group) was analyzed using linear discriminant analysis (LDA) effect size (LEfSe) on MicrobiomeAnalyst according to [21]. KEG Modules and KO terms with log LDA score higher than 2 and adjusted *p* < 0.05 for false discovery rate were considered significant.

## 3. Results

### 3.1. Diet Intake and Digestibility

There were no episodes of food refusal, vomiting, or diarrhea throughout the study, even during the abrupt dietary change. Dogs ate all the food offered, without leftovers. Dogs did not differ statistically in body weight and BCS during the study period or between treatments: control = 11.60 ± 1.49 kg and 6 ± 0.9 (day zero) to 11.80 ± 1.39 kg and 6 ± 0.6 (day 49) and probiotic = 11.51 ± 1.73 kg and 6 ± 0.9 (day zero) to 11.71 ± 1.40 kg and 6 ± 0.7 (day 49).

Food intake and nutrient intake differed between the LPF and HPF diets (*p* < 0.001), due to differences in ME content and chemical composition, but not between the control and probiotic groups (Table 2). There was a reduction in the ATTD of DM, OM, EE, and ME, and an increase in CP digestibility in the HPF diet, compared to the LPF diet, but the yeast supplementation did not statistically influence diet digestibility (Table 2).

Fecal scores did not differ statistically among days or between the control and probiotic groups—median control group = 4 (4;4) and median probiotic group = 4 (4;4). Wet fecal production and fecal DM did not differ statistically between control and probiotic groups, but the HPF diet resulted in greater fecal volume and lower fecal DM, than the LPF diet (*p* < 0.05, Table 2).

### 3.2. Fecal Fermentative Metabolites

There was a reduction in fecal pH and ammonia concentration after the diet change (*p* < 0.05, Figure 1). The probiotic group presented lower fecal ammonia concentration on day 23 in comparison to the control group (*p* < 0.05, Figure 1).

Except for putrescine, the fecal concentration of all biogenic amines decreased over time for both control and probiotic groups (*p* < 0.05, Table 3). Moreover, the probiotic group presented lower fecal concentration of spermidine regardless of the day, in relation to the control group (*p* < 0.05). The probiotic group also presented lower fecal concentration of total biogenic amines on days 21 and 49 than the control group (*p* < 0.05, Figure 1).

Fecal concentration of indole and p-cresol decreased over time (*p* < 0.05, Table 3), while the probiotic group presented lower fecal concentrations of indole, phenol, and p-cresol than the control group, regardless of the day (*p* < 0.05, Table 3).

In general, the LPF diet resulted in greater fecal concentrations of SCFA and BCFA on day 21, than the HPF diet on days 23 to 49 (Table 4). The probiotic group presented a higher fecal concentration of butyrate, isobutyrate, and total BCFA, regardless of the day (*p* < 0.05, Table 4).

### 3.3. Fecal Microbiome and Functional Profiling

#### 3.3.1. Dysbiosis Index and qPCR

Dogs fed the LPF diet presented a dysbiosis index considered equivocal (between 0 and 2, which is classified neither as diseased nor healthy), while after the transition to the HPF diet, those values decreased (lower than 0 on days 35 and 49) (*p* < 0.05, Figure 2). In addition, dogs from the probiotic group presented a lower dysbiosis index than the control group regardless of the day (*p* < 0.05, Figure 2).

The fecal *Turicibacter* concentration (logDNA) was reduced on day 23 (*p* < 0.05) but increased again on days 35 and 49 in both control and probiotic groups (Table 5). Dogs from the probiotic group presented an overall greater fecal concentration of *Turicibacter* than the control group regardless of the day (*p* < 0.05). The fecal concentration of *Fusobacterium* increased, while the fecal concentration of *C. hiranonis* and *Streptococcus* decreased on day 49 in both the control and probiotic groups (*p* < 0.05, Table 5). The probiotic group presented a lower fecal concentration of *E. coli* regardless of the day (*p* < 0.05) and a trend to lower *Streptococcus* (*p* = 0.098) than the control group (Table 5). Most of the bacteria quantified by qPCR were within the reference intervals [8], except for *Fusobacterium* (lower on days 0 to 35), *Turicibacter* (greater on all days), and *Streptococcus* (greater on days 21 and 23).

#### 3.3.2. Relative Microbial Abundance and Diversity Indexes

Relative to total eucaryote species identified in the feces, the probiotic group presented 96.3 ± 1.02% relative abundance of *Saccharomyces cerevisiae* on days 21 to 49, without differences among these days, while the control group presented 0 ± 0% on all days. This confirms that the yeast supplementation was detectable in the probiotic group, without cross contamination to the control group.

The main bacterial phyla identified were Firmicutes and Actinobacteria. The phylum Firmicutes reduced while the phylum Actinobacteria increased after the dietary change (*p* < 0.05). The probiotic group presented with lower Firmicutes and higher Actinobacteria relative abundances on days 35 and 49, than the control group (*p* < 0.05, Figure 3).

The main bacterial genera from the Firmicutes phylum were *Lactobacillus* and *Streptococcus*, while the main bacterial genera from the Actinobacteria phylum were *Collinsella* and *Bifidobacterium.* In addition, a change in the relative abundance of 15 genera over time was observed with the dietary change (*p* < 0.05, Appendix A). The main genera found in the feces of dogs fed the LPF diet (day 21) were *Streptococcus, Lactobacillus*, and *Collinsella*, while the main genera found after consumption of the HPF diet were *Collinsella, Bifidobacterium*, and *Blautia* (day 49). When comparing the effects of probiotic supplementation, there was a higher relative abundance (*p* < 0.05) of *Bifidobacterium* (days 35 and 49) and *Turicibacter* and a lower relative abundance of *Lactobacillus* for the probiotic group in comparison to the control group.

The relative abundance of the main bacterial species present in the feces of dogs is shown in Appendix A. *Bifidobacterium pseudolongum* was the most abundant species of its genus and presented with greater relative abundance on days 35 and 49 in the probiotic group (*p* < 0.05). While *Lactobacillus animalis* was the most abundant species of its genus and presented greater relative abundance in the control group (*p* < 0.05), regardless of the experimental day. Other predominant species were *Streptococcus lutetiensis* (greater abundance on days 0, 21, and 23), *Collinsella intestinalis* (lower abundance only on day 21)*, Clostridium hiranonis* (greater abundance in the probiotic group and on day 35)*, Turicibacter sanguinis* (greater abundance in the probiotic group and on days 0, 35, and 49)*,* and *Blautia hansenii* (greater abundance on day 0).

The dietary change influenced microbiome alpha diversity indexes (*p* < 0.05), with a reduction on day 21 followed by an increase on day 23—after the dietary change—in the control and probiotic groups. There was no difference in the microbiome diversity and richness between control and probiotic groups (Table 6).

Although the lack of difference in the microbiome diversity and richness between the treatments at each time point, the beta diversity analysis demonstrated that the dietary supplementation of the probiotic resulted in modulation of the intestinal microbiota on day 49 (*p* < 0.05) and a trend (*p* = 0.091) to change the microbiota on day 21 in comparison to the control group. In addition, the LPF (day 21) and HPF (day 49) diets also resulted in different microbiota profiles (*p* < 0.05, Figure 4).

### 3.4. Functional Genes

The LPF (day 21) and HPF (day 49) diets resulted in different functional genes expression (*p* < 0.05). Control and probiotic groups also presented different (day 49, fed the HPF diet, *p* < 0.05) or a trend to different (day 21, fed the LPF diet, *p* = 0.081) microbial functional genes expression (Figure 4).

When evaluating only the effects of the diets, an enrichment in some sugar transport system modules (M00197 and M00196), ornithine (M00028) and histidine (M00026) biosynthesis, and Shikimate pathway (M00022) were observed in the feces of dogs fed the HPF diet (*p* < 0.05), while dogs fed the LPF diet presented enrichment mainly in phosphoenolpyruvate (PEP)-dependent phosphotransferase systems (PTS) (M00268, M00269, M00273, M00270, M00275, and M00276), glycolysis (M00001 and M00002), leucine degradation (M00036), biosynthesis of some amino acids (M00525, M00018, M00015, and M00021) and lipids (M00082, M00083, and M00089), spermidine/putrescine transport system (M00299), methanogenesis (M00357), osmoprotectant systems (M00209 and M00208), and two-component regulatory systems (M00459 and M00434, *p* < 0.05, Figure 5). Besides the enrichment in these KEGG Modules, the LPF diet also resulted in the upregulation of genes (KO terms and pathways) related to microbial metabolism in diverse environments (ko01120), carbon metabolism (ko01200), starch and sucrose metabolism (ko00500), amino sugar and nucleotide sugar metabolism (ko00520), polysaccharide transporter, PST family (ko03328), pyruvate metabolism (ko00620), ABC transporters (ko02010), HIF-1 signaling pathway (ko04066), propanoate and butanoate metabolism (ko00640 and ko00650), resistance to antibiotics (ko01501, ko01503, ko00687, ko10823, ko12555, ko15583, and ko14205), *E. coli* pathogenicity (ko02026, ko05130, ko00134, ko00694, and ko00975), and *Salmonella* and *Staphylococcus aureus* infection (ko05132, ko00134, ko14215, and ko05150) (Appendix A). 

Considering the comparison between the control and probiotic groups fed the LPF diet (day 21, Figure 6), we observed enrichment in most of the modules and KO orthologs previously described to the LPF diet only in the control group (*p* < 0.05). This demonstrates that the probiotic supplementation presented an effect to control the expression of these genes. In addition, the control group also presented enrichment (*p* < 0.05) in Modules related to isoprenoid biosynthesis (M00364, M00095, and M00366), autoinducer-2 (AI-2) system (M00219), cytochrome d ubiquinol oxidase (M00153), and amino acids and dipeptide transport systems (M00435, M00236, M00234, and M00566). Regarding KO terms and pathways, the control group also presented enrichment in genes related to homologous DNA recombination (ko03440), sulfur metabolism (ko00920), branched-chain amino acid:cation transporter, LIVCS family (ko03311), metabolism of xenobiotics by cytochrome p450 (ko00980), naphthalene degradation (ko00626), and degradation of aromatic compounds (ko01220) on day 21 (Appendix A).

The differences found in KEGG Modules between the control and probiotic groups fed the HPF diet (day 49, Figure 6) were enrichment in some of the two-component transport systems (M00479 and M00481), bacterial proteasome (M00342), putative ABC transport system (M00211), manganese/zinc/iron transport system (M00319), and beta-oxidation (M00086) in the probiotic group (*p* < 0.05). Regarding KO terms, besides the pathways directly related to the modules, the probiotic group also presented enrichment in genes related to starch and sucrose metabolism (ko00500), biosynthesis of amino acids (ko01230), peptidoglycan biosynthesis (ko00550), quorum sensing (ko02024), homologous recombination (ko03440), nitrogen metabolism (ko00910), lipoarabinomannan (LAM) biosynthesis (ko00571), inositol phosphate metabolism (ko00562), peroxisome (ko04146), sphingolipid metabolism (ko00600), pyruvate metabolism (ko00620), thermogenesis (ko04714), adipocytokine signaling pathway (ko04920), selenocompound metabolism (ko00450), and nucleotide excision repair (ko03420). The main modules upregulation found in the control group were lipopolysaccharide export system (M00320), PTS systems (M00268 and MM00270), transport systems (M00254 and M00212), formaldehyde assimilation (M00344), and VicK-VicR (cell wall metabolism) two-component regulatory system (M00459), while the main KO terms enriched in the control group were choline trimethylamine-lyase (ko05349), beta-glucosidase (ko01223), and ATP-dependent helicase/nuclease subunit A (ko16898) on day 49 (Appendix A).

## 4. Discussion

A possible challenge in animal studies with probiotics is to avoid cross-contamination between the experimental groups. This issue was evaluated through shotgun sequencing analysis, confirming that the yeast supplementation was effective in the probiotic group, without contamination of the control group, in which *S. cerevisiae* was not identified in the feces.

The hypothesis of the study was that animals in the probiotic group would display a greater ability to adapt to the new diet composition, mainly due to a more stable gut microbiome. This would be expected considering that the yeast may act as a probiotic, beneficially affecting the gut environment and microbiome function. Surprisingly, based on the results found, it seems that the gastrointestinal tract of dogs can have the capability to quickly adapt to a new diet. Similar results were reported by [22], who described that in general fecal consistency and most fecal metabolites and microbiota changed in a few days (2–6 days) after an abrupt dietary change in dogs, with complete stabilization after two weeks. However, these findings must be extrapolated carefully to pet dogs, considering that laboratory dogs may be more used to dietary changes and there are a lot of individual factors that may affect the gastrointestinal response to dietary change. The fact that the dogs in the present study did not present adverse reactions to the dietary change may also be attributed to the nutritional characteristics of the HPF diet used, which has highly digestible protein and low concentrations of fermentable fiber.

Considering that most of the digestion process takes place in the upper gastrointestinal tract of dogs, there would not necessarily be a significant impact of yeast supplementation on the digestibility of the diets, but an effect on the microbiota modulation and fermentative metabolites in the colon. In this sense, the lack of effect on digestibility of DM, CP, and EE has already been reported by other authors who evaluated the dietary inclusion of yeast products in dogs [4,23].

Regarding fecal characteristics, the HPF diet resulted in the production of softer and bulky feces than the LPF diet, due to its higher fiber [24,25] and lower ash concentrations. However, both diets resulted in well-formed feces, without episodes of diarrhea or constipation, even after the abrupt dietary change. This may be one of the reasons that we did not observe any effect of the yeast supplementation on fecal consistency.

Fecal pH can be considered a biomarker of intestinal microbiota fermentative activity, with the highest pH corresponding to proteolytic metabolism [26]. Therefore, the reduction in fecal pH after the dietary change and yeast supplementation may be explained due to the lower fecal concentrations of ammonia and total biogenic amines observed. Besides, the reduction in the gut pH may contribute to intestinal functionality by inhibiting the proliferation of potentially pathogenic bacteria [27,28].

Interestingly, the reduction in the fecal pH in dogs fed the HPF diet was followed by a reduction in the fecal concentration of SCFA and not by their increase, as could be expected. However, the HPF diet used had high digestibility and more IF (26.1%) than SF (1.8%) concentration along with fiber sources with low (cellulose, linseed) to moderate (soy hulls) fermentability [25].

Among SCFA, butyrate is recognized for its important role in reducing inflammation and regulating the epithelial barrier function [29]. In addition, it represents the main energy source for intestinal epithelial cells [28]. In the current study, yeast probiotic supplementation resulted in higher fecal butyrate concentration, regardless of the diet. Similar results were found in nursery pigs receiving live *S. cerevisiae* in their diet [30].

Regarding amino acid fermentation catabolites, in high concentrations, some of them can have negative effects on intestinal functionality. In excess, these catabolites may be toxic to the intestinal mucosa and may favor the survival of potentially pathogenic bacteria [31]. The fecal concentration of indole, phenols, total biogenic amines, ammonia, and p-cresol was lower in the probiotic group compared to the control group. Thus, these results may indicate that the supplementation of live yeast may have beneficial effects on the intestinal functionality of dogs. On the other hand, although studies report the toxic effects of higher concentrations of nitrogen fermentation products on colonocytes [31,32], the threshold between the functional and toxic concentrations of these compounds for dogs is not yet known.

In addition to the alteration in the production of some fermentative metabolites, the main changes identified in bacterial phyla were a reduction in Firmicutes (mainly of the *Lactobacillus* genus) and an increase in Actinobacteria (mainly of the *Bifidobacterium* genus) after the change to the HPF diet associated with the probiotic supplementation. The interaction observed between diet × probiotic supplementation indicates that the live *S. cerevisiae* (Actisaf Sc50 PET product) evaluated seems to have greater capability to modulate the gut microbiota of dogs when higher fiber concentration is present in the diet. This hypothesis is also supported by the beta diversity results, which demonstrated a higher differentiation of the microbial profile between dogs from the control and probiotic group on day 49 (HPF diet), than on day 21 (LPF diet). In nursery pigs, [30] also described an increase in the relative abundance of Actinobacteria phylum and *Bifidobacterium* genus in animals receiving live *S. cerevisiae*. According to the authors [30], these results might be indicative of competition between Bifidobacteria and Lactobacilli over yeast-derived changes in the profile of hindgut metabolites.

Studies indicate that the *Bifidobacterium* genus may have beneficial effects on inflammatory and immune-driven diseases via the regulation of specific immune cells and cellular networks, including cytokines, controlling the inflammatory process [33]. Furthermore, *Bifidobacterium* is also commonly decreased in fecal and mucosal samples from human and cat patients with inflammatory bowel disease [34,35].

Besides the shifts observed in *Bifidobacterium* and *Lactobacillus*, dogs from the probiotic group also presented alterations in other bacterial genera, such as an increase in *Turicibacter* (important to serotonin metabolism in the gut) and *Clostridium hiranonis* (conversion of primary bile acids to secondary bile acids) and a reduction in *E. coli*, that is a potential pathogen [8,36,37,38,39]. The reduction in *E. coli* counts was also observed in the feces of dogs receiving dietary supplementation of live *S. cerevisiae* [4].

These changes in *Turicibacter, C. hiranonis*, and *E. coli* are in accordance with the results of dysbiosis index, indicating an improvement in the gut eubiosis in dogs from the probiotic group and in dogs fed the HPF diet. Many diseases, systemic or local, are associated directly or indirectly with gut dysbiosis. A balanced gut microbiome exerts a positive influence on the overall health of the host by modulating the immune system, defending against potential intestinal pathogens, and providing vitamins and important metabolites [39].

The mechanisms of action of live yeast in the gastrointestinal tract of non-ruminant animals are not well established. They are usually attributed to the stimulation of brush-edge disaccharidases; competition for adhesion sites against potential pathogens; stimulation of nonspecific immunity; neutralization of toxins; and direct effect against potentially pathogenic microorganisms [40,41]. Many of these mechanisms are influenced directly by yeast cell wall components, such as mannans and beta-glucans. One of the main mechanisms of action of mannans is the elimination of bacteria with pathogenic potential that present type-1 fimbriae, such as some strains of *E. coli* and *Salmonella*, preventing their adhesion and colonization to the host mucosa. This may also contribute to the establishment of beneficial bacteria [42,43].

In the present study, the functional analysis of the gut microbiome identified the upregulation of genes related to microbial virulence factors, such as lipopolysaccharide export system and two-component regulatory system VicRK only in the control group, indicating a possible protective effect of the probiotic against potential pathogenic bacteria. Lipopolysaccharide is the major component of the outer membrane of Gram-negative bacteria, such as *E. coli*, which is involved in toxicity, pathogenicity, and antimicrobial resistance. While the two-component regulatory system VicRK is important to the virulence of some Gram-positive bacteria, such as *Streptococcus* sp. [44]. Besides these virulence factors, dogs from the control group also presented enrichment in genes related to DNA repair and recombinant proteins, which are also identified in non-probiotic and pathogenic microorganisms [45].

Regarding functional genes related to fermentative metabolites, the control group fed the LPF diet presented upregulation of genes related to spermidine/putrescine transport, methanogenesis, and degradation of aromatic compounds. These results are in accordance with the higher fecal concentration of putrefactive compounds in the feces of dogs on day 21. In addition, the higher production of fermentative metabolites on day 21 may explain the upregulation of osmoprotectant systems, which are activated when bacteria are under osmotic stress mainly due to the higher concentration of nitrogen-fermentative metabolites [46].

The functional genes that were upregulated in the probiotic group fed the HPF diet were also described in other studies evaluating probiotics in relation to non-probiotic species [45,47]. Our findings and those from [45,47] observed that in general probiotic species resulted in higher gene expression related to lipid biosynthesis, carbohydrate metabolism and uptake, biosynthesis of secondary metabolites, and essential amino acid biosynthesis. These functions are possibly related to benefits to gastrointestinal functionality, such as the synthesis of important metabolites to gut homeostasis. However, considering the complexity of interactions that occur in the gastrointestinal tract and the lack of the establishment of thresholds between the beneficial and harmful concentrations of gut metabolites, it is important that further metagenomic studies be conducted to better elucidate these factors.

The main limitation of the present study was the limited number of dogs used, considering that microbiome and functional analysis data are highly variable. Thus, it is important that future studies validate these results.

## 5. Conclusions

Our results suggest that live *S. cerevisiae* supplementation may improve indicatives of gastrointestinal functionality in dogs, by reducing the fecal concentration of some nitrogen fermentative catabolites and increasing the overall fecal concentration of butyrate, regardless of the diet. In addition, the live yeast supplementation also seems to modulate the gut microbiota and its functions, favoring eubiosis and gut homeostasis and reducing potential pathogens, such as *E. coli.* Considering these results, the supplementation of *S. cerevisiae* as a probiotic in commercial dog food may contribute to health benefits for these animals. However, it is important that future studies with a higher number of repetitions validate these potential effects.

## Figures and Tables

**Figure 1 microorganisms-11-00506-f001:**
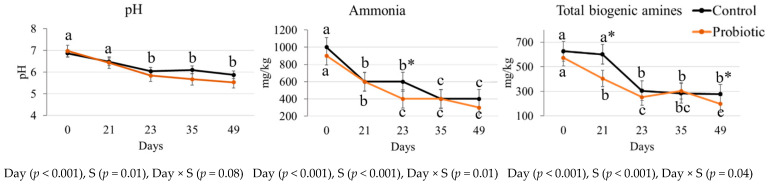
Fecal pH and concentrations of ammonia and total biogenic amines in dogs fed a lower protein and fiber diet (day 21) transitioned to a higher protein and fiber diet (days 23 to 49) without (control) or with (probiotic) yeast supplementation (S). ^a,b,c^ Different superscript letters near control and probiotic groups for ammonia and total biogenic amines indicate differences among days by the LSD Bonferroni test in each group (*p* < 0.05). When *p* > 0.05 for interaction, the letters indicate difference for day for both groups (fecal pH). * Indicates difference between control and probiotic groups at the time point.

**Figure 2 microorganisms-11-00506-f002:**
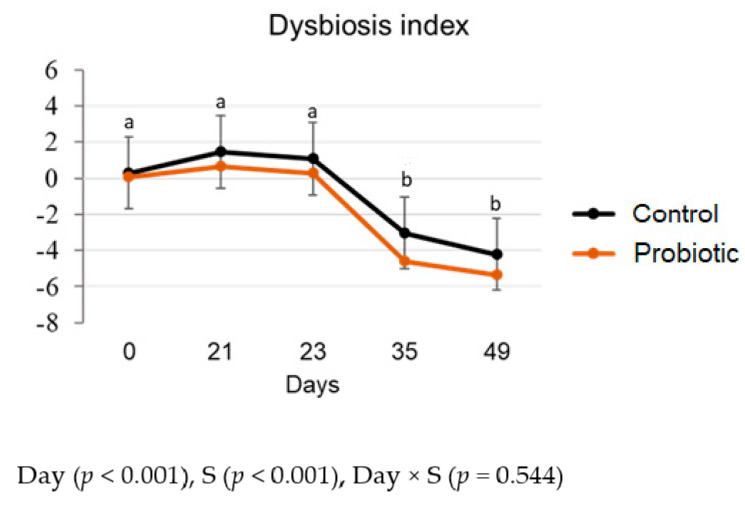
Dysbiosis index of dogs fed a lower protein and fiber diet (day 21) transitioned to a higher protein and fiber diet (days 23 to 49) without (control) or with (probiotic) yeast supplementation (S). ^a,b^ Different superscript letters indicate differences among days by the LSD Bonferroni test (*p* < 0.05) for both control and probiotic groups.

**Figure 3 microorganisms-11-00506-f003:**
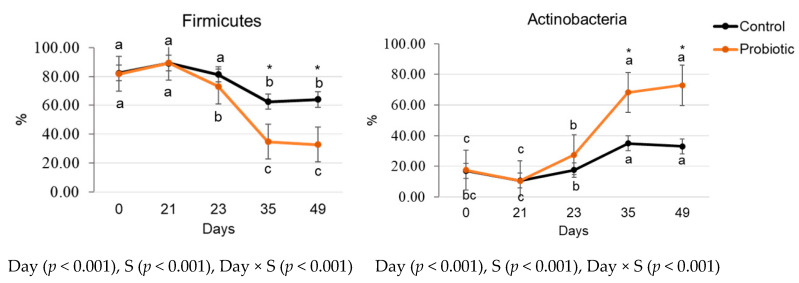
Relative abundance of the main bacterial phyla in feces of dogs fed a lower protein and fiber diet (day 21) transitioned to a higher protein and fiber diet (days 23 to 49) without (control) or with (probiotic) yeast supplementation (S).^a,b,c^ Different superscript letters near control and probiotic groups indicate differences among days by the LSD Bonferroni test in each group (*p* < 0.05). * Indicates difference between the control and probiotic groups at the time point (*p* < 0.05).

**Figure 4 microorganisms-11-00506-f004:**
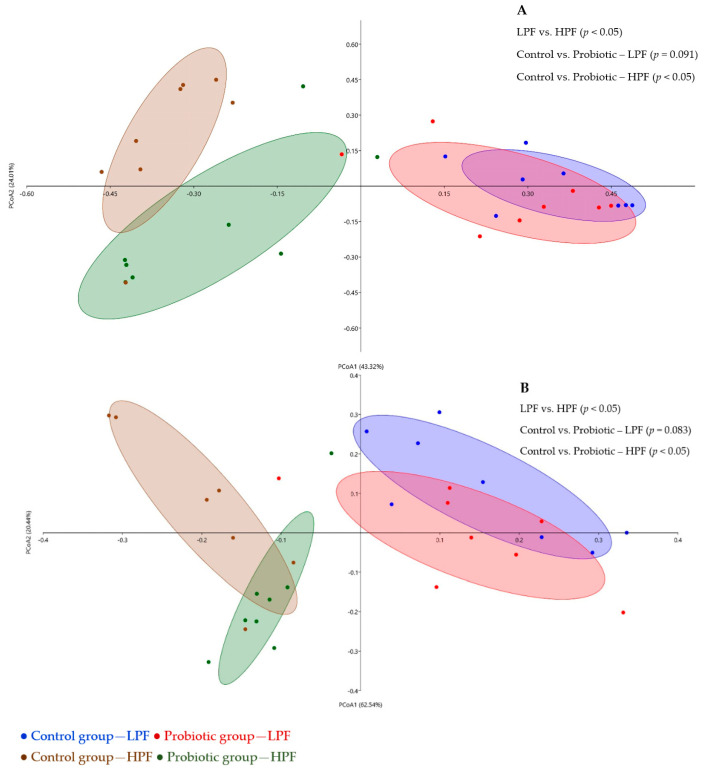
Principal coordinate analysis (PCoA) of the composition of bacterial communities (**A**) and its functional genes (**B**) of control and probiotic groups fed the lower protein and fiber diet (LPF, day 21) and the higher protein and fiber diet (HPF, day 49). The figure was elaborated using the Bray–Curtis dissimilarity method and represents the degree of difference among samples. Each square/dot represents an animal. *p*-values between comparisons are from ANOSIM analysis.

**Figure 5 microorganisms-11-00506-f005:**
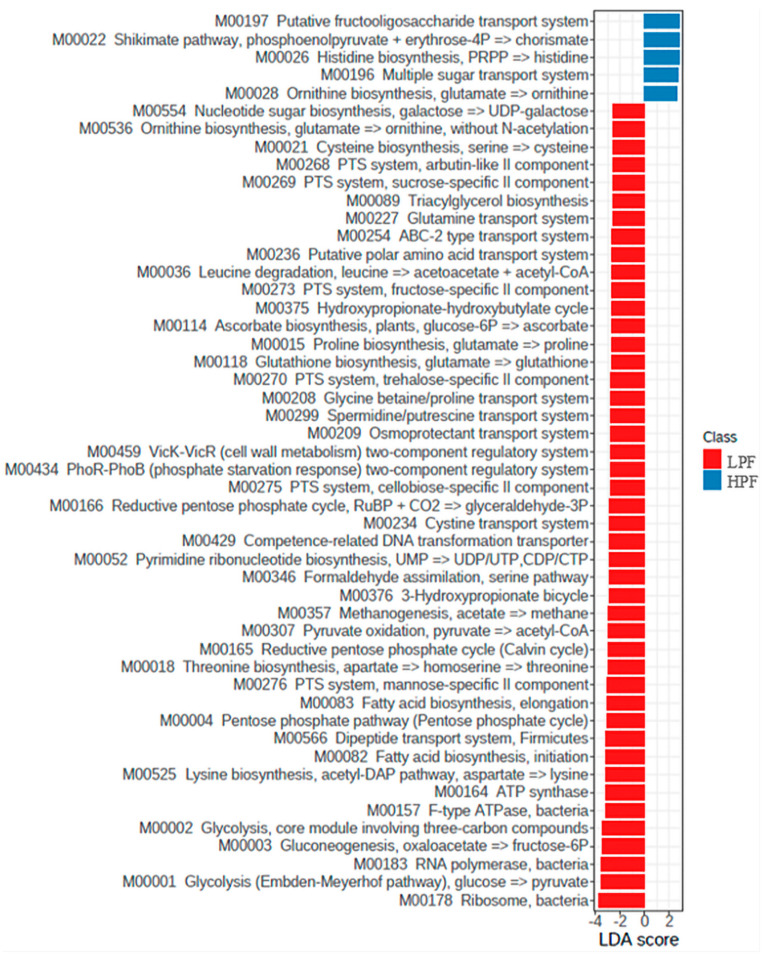
LDA score of KEGG Modules that statistically differ (*p* < 0.05) between the lower protein and fiber diet (LPF, day 21) and the higher protein and fiber diet (HPF, day 49).

**Figure 6 microorganisms-11-00506-f006:**
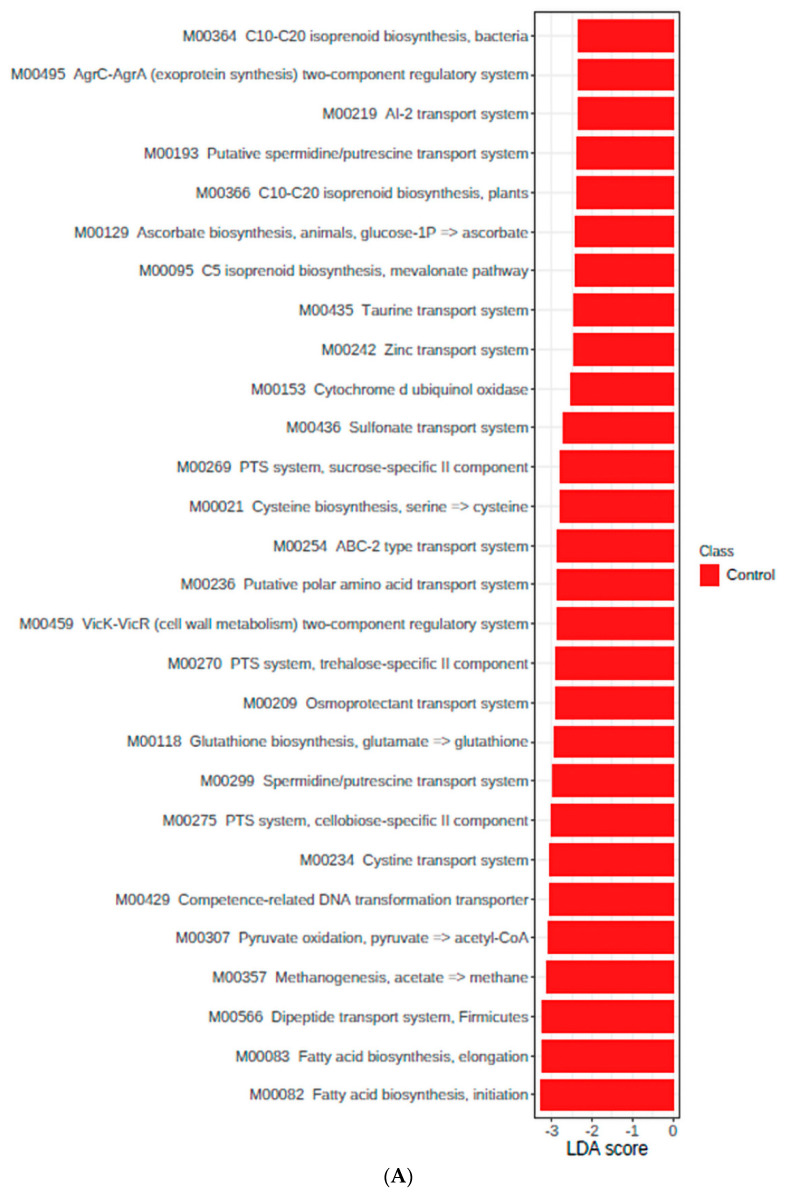
LDA score of modules that statistically differ (*p* < 0.05) between the control and probiotic groups fed the lower protein and fiber diet—day 21 (**A**) and the higher protein and fiber diet—day 49 (**B**).

**Table 1 microorganisms-11-00506-t001:** Analyzed chemical composition (%, dry matter basis) of the lower protein and fiber diet (LPF) and higher protein and fiber diet (HPF).

Item	LPF	HPF
Ash	10.44	5.81
Crude Protein	20.42	27.52
Ether Extract in Acid Hydrolysis	10.25	8.02
Total Dietary Fiber	6.10	27.20
Insoluble Fiber	4.89	26.10
Soluble Fiber	1.21	1.80
Calcium	1.68	0.76
Phosphorus	1.42	0.63
Metabolizable Energy (kcal/kg) *	3.600	2.912

*: Value presented in the diet label. Main ingredients LPF: Corn, poultry by-products meal, meat and bone meal, soybean meal, chicken hydrolysate, poultry fat, vitamins, and minerals. HPF: Corn, poultry by-products meal, corn gluten meal, soy hulls, cellulose, chicken and swine hydrolysate, poultry fat, soybean oil, linseed, vitamins, and minerals.

**Table 2 microorganisms-11-00506-t002:** Means in dry matter (DM) and nutrient intake, coefficients of total tract apparent digestibility (ATTD, %) of nutrients, metabolizable energy (ME, kcal/kg), and fecal characteristics of dogs fed a lower protein and fiber (LPF) or a higher protein and fiber (HPF) diet (D) without (control) or with (probiotic) yeast supplementation (S).

Item	D	Supplementation (S)		SEM	*p*-Values		
		Control	Probiotic		D	S	D × S
Intake (g DM basis/dog/day)							
DM	LPF	181.6	181.7	3.66	<0.001	0.302	0.291
	HPF	244.0	233.5				
CP	LPF	37.1	37.1	2.67	<0.001	0.110	0.110
	HPF	68.0	64.2				
EE	LPF	18.6	18.6	0.24	0.122	0.245	0.235
	HPF	19.8	18.8				
IF	LPF	8.9	8.9	0.97	<0.001	0.092	0.092
	HPF	63.7	60.9				
SF	LPF	2.2	2.2	0.47	<0.001	0.078	0.073
	HPF	4.4	4.2				
ME	LPF	708.4	713.4	42.37	<0.001	0.296	0.201
	HPF	858.2	811.4				
ATTD							
DM	LPF	82.0	82.3	0.88	<0.001	0.758	0.984
	HPF	73.6	73.9				
OM	LPF	87.3	87.4	1.19	<0.001	0.818	0.885
	HPF	74.5	74.8				
CP	LPF	83.2	83.7	0.55	<0.001	0.899	0.682
	HPF	87.5	87.3				
EE	LPF	89.9	90.2	0.39	<0.001	0.657	0.947
	HPF	86.7	87.0				
ME	LPF	3899.5	3929.2	28.01	<0.001	0.626	0.644
	HPF	3472.1	3472.9				
Fecal characteristics							
Wet feces	LPF	89.6	85.6	8.89	<0.001	0.298	0.646
	HPF	184.8	174.7				
DM (%)	LPF	36.9	37.4	0.56	0.046	0.855	0.828
	HPF	34.9	34.9				

Dry matter: DM; Crude protein: CP; Ether extract in acid hydrolysis: EE; Insoluble fiber: IF; Soluble fiber: SF; Metabolizable energy: ME; Organic matter: OM. SEM: Standard error of mean. Wet feces (g/dog/day).

**Table 3 microorganisms-11-00506-t003:** Means of the fecal concentration (mg/kg dry matter) of biogenic ammines and % of pic areas of indole, phenol, and p-cresol of dogs fed a lower protein and fiber diet (day 21) transitioned to a higher protein and fiber diet (days 23 to 49) without (C) or with (P) yeast supplementation (S).

Item	S	Day					SEM	*p*-Values		
		0	21	23	35	49		Day	S	Day × S
Spermidine	C	148.4 ^a^	94.9 ^b^	119.8 ^b^	99.6 ^b^	104.1 ^b^	8.2	<0.001	0.041	0.183
	P	135.8 ^a^	94.8 ^b^	103.4 ^b^	91.2 ^b^	76.1 ^b^	8.4			
Spermine	C	185.0 ^a^	36.8 ^b^	49.1 ^b^	51.5 ^b^	35.5 ^b^	24.1	<0.001	0.841	0.874
	P	178.7 ^a^	33.5 ^b^	44.6 ^b^	59.3 ^b^	37.3 ^b^	23.1			
Putrescine	C	38.0 ^b^	143.8 ^a^*	81.5 ^ab^	82.9 ^ab^	116.9 ^a^*	15.1	<0.001	0.222	0.022
	P	52.8 ^a^	44.3 ^a^*	63.0 ^a^	100.8 ^a^	79.8 ^a^*	8.5			
Tyramine	C	254.8 ^a^	326.2 ^a^	53.6 ^b^	49.8 ^b^	19.4 ^b^	52.8	<0.001	0.173	0.872
	P	205.7 ^a^	232.0 ^a^	41.8 ^b^	52.7 ^b^	4.5 ^b^	39.2			
Indole	C	2.3 ^a^	2.1 ^a^	2.3 ^a^	1.5 ^b^	1.6 ^b^	0.1	0.021	<0.001	0.081
	P	2.0 ^a^	1.9 ^a^	0.3 ^a^	0.6 ^b^	0.6 ^b^	0.3			
Phenols	C	0.89	1.05	1.89	1.83	1.65	0.3	0.476	0.040	0.069
	P	1.75	0.61	0.94	0.98	0.52	0.2			
p-cresol	C	0.85 ^a^	1.27 ^a^	0.77 ^b^	0.62 ^b^	1.61 ^a^	0.2	0.015	<0.001	0.140
	P	1.40 ^a^	0.93 ^a^	0.46 ^b^	0.54 ^b^	0.36 ^a^	0.2			

^a,b^ Different superscript letters indicate differences among days by the LSD Bonferroni test in each group (*p* < 0.05 for Day × S) or for both groups (*p* > 0.05 for Day × S). * Indicate difference between control and probiotic groups at the time point. SEM: standard error of the mean.

**Table 4 microorganisms-11-00506-t004:** Means of fecal concentration (µmol/g dry matter) of short-chain fatty acids (SCFA) and branched-chain fatty acids (BCFA) of dogs fed a lower protein and fiber diet (day 21) transitioned to a higher protein and fiber diet (days 23 to 49) without (C) or with (P) yeast supplementation (S).

Item	S	Day					SEM	*p*-Values		
		0	21	23	35	49		Day	S	Day × S
SCFA										
Acetate	C	156.8 ^b^	206.8 ^a^	131.9 ^b^	138.7 ^b^	133.5 ^b^	11.8	<0.001	0.824	0.367
	P	140.3 ^b^	200.7 ^a^	155.4 ^b^	155.8 ^b^	123.7 ^b^	10.8			
Propionate	C	50.3 ^b^	103.4 ^a^	58.5 ^b^	49.1 ^b^	46.7 ^b^	9.0	<0.001	0.354	0.089
	P	46.6 ^b^	85.8 ^a^	61.7 ^b^	58.6 ^b^	39.6 ^b^	6.3			
Butyrate	C	11.1 ^a^	11.3 ^a^	7.4 ^b^	5.9 ^b^	5.6 ^b^	1.0	<0.001	0.040	0.149
	P	13.4 ^a^	12.3 ^a^	7.4 ^b^	8.6 ^b^	6.0 ^b^	2.0			
Valerate	C	3.4 ^a^	2.6 ^ab^	2.0 ^b^	2.0 ^b^	1.9 ^b^	0.2	<0.001	0.105	0.459
	P	3.2 ^a^	2.9 ^ab^	2.2 ^b^	2.3 ^b^	1.9 ^b^	0.2			
Total SCFA	C	221.6 ^b^	324.1 ^a^	199.8 ^b^	195.7 ^b^	187.7 ^b^	21.3	<0.001	0.990	0.329
	P	204.5 ^b^	301.7 ^a^	226.7 ^b^	225.3 ^b^	171.2 ^b^	18.1			
BCFA										
Isovalerate	C	1.0 ^ab^	1.2 ^a^	1.1 ^a^	0.9 ^ab^	0.9 ^b^	0.01	<0.001	0.850	0.880
	P	1.0 ^ab^	1.3 ^a^	1.1 ^a^	1.0 ^ab^	0.8 ^b^	0.01			
Isobutyrate	C	2.2 ^a^	2.0 ^a^	1.4 ^b^	1.3 ^b^	1.5 ^b^	0.1	<0.001	0.048	0.647
	P	2.3 ^a^	2.2 ^a^	1.7 ^b^	1.6 ^b^	1.5 ^b^	0.1			
Total BCFA	C	3.2 ^a^	3.3 ^a^	2.5 ^b^	2.3 ^b^	2.4 ^b^	0.2	<0.001	0.028	0.338
	P	3.2 ^a^	3.5 ^a^	2.8 ^b^	2.7 ^b^	2.3 ^b^	0.2			

^a,b^ Different superscript letters indicate differences among days by the LSD Bonferroni test for both groups (*p* < 0.05). SEM: standard error of the mean.

**Table 5 microorganisms-11-00506-t005:** Means of fecal concentration (logDNA) of selected bacterial groups of dogs fed a lower protein and fiber diet (day 21) transitioned to a higher protein and fiber diet (days 23 to 49) without (C) or with (P) yeast supplementation (S).

Item	S	Day					SEM	*p*-Values		
		0	21	23	35	49		Day	S	Day × S
*Faecalibacterium*	C	4.5	4.2	4.5	4.5	4.5	0.47	0.544	0.929	0.971
	P	4.3	4.3	4.7	4.6	4.5	0.06			
*Fusobacterium*	C	6.6 ^b^	6.5 ^b^	6.7 ^b^	6.7 ^b^	7.3 ^a^	0.11	<0.001	0.238	0.757
	P	6.6 ^b^	6.6 ^b^	6.6 ^b^	6.9 ^b^	7.5 ^a^	0.16			
*Blautia*	C	10.4 ^a^	9.8 ^b^	10.1 ^ab^	10.0 ^ab^	10.2 ^a^	0.08	<0.001	0.163	0.436
	P	10.3 ^a^	10.1 ^b^	10.1 ^ab^	10.2 ^ab^	10.3 ^a^	0.04			
*Turicibacter*	C	8.7 ^a^	8.6 ^a^	8.2 ^b^	8.6 ^a^	8.8 ^a^	0.08	0.017	0.028	0.952
	P	9.1 ^a^	9.0 ^a^	8.4 ^b^	8.8 ^a^	8.9 ^a^	0.09			
*C. hiranonis*	C	6.1 ^a^	5.9 ^a^	6.0 ^a^	5.8 ^ab^	5.7 ^b^	0.06	0.008	0.311	0.711
	P	6.1 ^a^	6.1 ^a^	5.9 ^a^	5.9 ^ab^	5.8 ^b^	0.06			
*Streptococcus*	C	7.8 ^a^	8.6 ^a^	8.1 ^a^	4.9 ^b^	3.7 ^b^	0.82	<0.001	0.098	0.276
	P	7.8 ^a^	8.6 ^a^	7.8 ^a^	3.9 ^b^	3.5 ^b^	0.91			
*E. coli*	C	4.3 ^a^	4.2 ^ab^	4.7 ^a^	3.4 ^b^	4.8 ^a^	0.20	0.030	0.043	0.256
	P	4.2 ^a^	3.7 ^ab^	3.7 ^a^	3.7 ^b^	3.7 ^a^	0.11			
Total bacteria	C	10.9 ^a^	11.0 ^a^	10.9 ^a^*	10.5 ^b^	10.7 ^ab^	0.08	<0.001	0.520	0.030
	P	11.0 ^a^	10.9 ^a^	10.6 ^a^*	10.7 ^a^	10.9 ^a^	0.05			

^a,b^ Different superscript letters indicate differences among days by the LSD Bonferroni test in each group (*p* < 0.05 for Day × S) or for both groups (*p* > 0.05 for Day × S). * Indicate differences between the control and probiotic groups at the time point. SEM: standard error of the mean.

**Table 6 microorganisms-11-00506-t006:** Means of alpha diversity indexes of dogs fed a lower protein and fiber diet (day 21) transitioned to a higher protein and fiber diet (days 23 to 49) without (C) or with (P) yeast supplementation (S).

Item	S	Day (D)					SEM	*p*-Values		
		0	21	23	35	49		D	S	D × S
Chao1	C	408.9 ^a^	332.7 ^b^	422.8 ^a^	389.2 ^ab^	407.2 ^a^	9.36	<0.001	0.521	0.208
	P	415.8 ^a^	356.2 ^b^	462.8 ^a^	396.6 ^ab^	384.9 ^a^	11.00			
OTUs	C	346.4 ^a^	255.1 ^b^	348.9 ^a^	344.1 ^a^	359.0 ^a^	10.77	<0.001	0.406	0.131
	P	356.4 ^a^	295.6 ^b^	393.8 ^a^	335.3 ^a^	328.3 ^a^	10.06			
Shannon	C	3.2 ^ab^	2.7 ^b^	3.3 ^a^	3.3 ^a^	3.8 ^a^	0.11	0.030	0.186	0.691
	P	3.3 ^ab^	3.0 ^b^	3.6 ^a^	3.6 ^a^	3.7 ^a^	0.07			

^a,b^ Different superscript letters indicate differences among days by the LSD Bonferroni test for both groups (*p* < 0.05). SEM: standard error of the mean.

## Data Availability

The essential data supporting the reported results are contained in this study and in the Appendix A. All other data can be made available on request to the corresponding author.

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
