# Peer review of "Effect of Yeast Saccharomyces cerevisiae as a Probiotic on Diet Digestibility, Fermentative Metabolites, and Composition and Functional Potential of the Fecal Microbiota of Dogs Submitted to an Abrupt Dietary Change"

_microorganisms, 2023, doi:10.3390/microorganisms11020506_

Round 1

Reviewer 1 Report

In this paper, the authors depicted the effects of live yeast supplementation in 8 out of 16 dogs by assessing the digestibility of nutrients and energy, intestinal fermentative products, the fecal microbiota, and functional genes.

Although the results seem interesting and novel, the conclusions drawn by the authors are significantly biased by the low number of animals they included. It is widely acknowledged that for genomic and gut microbial composition, it is necessary to deal with more robust numbers of samples...

For this reason, I recommend rejecting the paper unless the authors provide more data on a wider animal number.

Moreover, I suggest removing the “p>0.05” throughout the text, specifying which ANOVA was performed, clarifying the sentence of line 100 (…according to [11] what? AOAC?) as well as whether it is a trial or a study (both terms are used in the text).

Author Response

Reviewer 1

In this paper, the authors depicted the effects of live yeast supplementation in 8 out of 16 dogs by assessing the digestibility of nutrients and energy, intestinal fermentative products, the fecal microbiota, and functional genes.

Although the results seem interesting and novel, the conclusions drawn by the authors are significantly biased by the low number of animals they included. It is widely acknowledged that for genomic and gut microbial composition, it is necessary to deal with more robust numbers of samples...

For this reason, I recommend rejecting the paper unless the authors provide more data on a wider animal number.

Moreover, I suggest removing the “p>0.05” throughout the text, specifying which ANOVA was performed, clarifying the sentence of line 100 (…according to [11] what? AOAC?) as well as whether it is a trial or a study (both terms are used in the text).

Au: Thank you very much for all your comments and suggestions. We improved the manuscript accordingly. Unfortunately, we could not use more animals because they are laboratory dogs, and we have a worldwide recommendation to reduce the number of animals used in research. More than a recommendation, here at our University, the Ethics Committee in Animal Use does not approve nutritional trials in dogs with more than 8 animals/treatment, unless they are household pets. The main advantage to use laboratory dogs in nutritional trials, especially the ones evaluating the gut microbiome, is that the conditions are highly controlled (dogs’ sex, age, body condition, environment, diet, health, management, fecal collection, etc), and thus the results are more consistent than when we conduct studies with household pets. In household pets we need a high number of animals, considering the higher variability of the data. Besides, in our study the statistical analysis found differences considering the number of repetitions and the variability of the data. The main problem with a low number of repetitions is the possible lack of statistical power to find differences among groups in some variables. But this is more acceptable (statistical error type II) than a false positive result (error type I). This is why we also corrected the p-values for false discovery rate in the microbiome and functional genes data. We include a paragraph to state the low number of dogs as a limitation of our study and encourage further studies to validate the results.

We removed “P>0.05” of the text.

The ANOVA used is described in the text. We used the function general linear model and included the effects of yeast supplementation and sex (fixed effects) and day of measurement (repeated measure) in the ANOVA.

On line 100, the analysis was conducted according to the AOAC (1995) methods. We followed the journal guidelines to present the in-text reference in this way.

We standardize throughout the text to the term “study”.

Reviewer 2 Report

The manuscript is nicely done and written. The study design is appropriate and apparently, the analyses were carefully performed.  I believe that the results are valuable for the scientific community and has significant scientific merit, as it will probably ignite many further studies in the near future. The literature is up to date and conclusions are justified by obtained results. I do not have any emerging concerns about the scientific aspects of the manuscript. However, the manuscript could be improved by making some changes as suggested as follows:

Line 68 – Please remove table 1 with ingredients of diets into the main text.

Line 98 – please change to 105oC.

Line 128 – please choose one version: either SHIMADZU or Shimadzu® (line 134)

Line 142 – please provide a software license if necessary.

Line 176 – KEGG was already abbreviated in line 49.

Line 176 – the first appearance of KO should be in line 49.

Line 184 – Bacteria’s phyla should be written in italics.

Line 581 – the authors should discuss the limitations of the study at the end of the chapter.

Line 586 – please add chapter with conclusions and future perspectives.

Author Response

Reviewer 2

The manuscript is nicely done and written. The study design is appropriate and apparently, the analyses were carefully performed.  I believe that the results are valuable for the scientific community and has significant scientific merit, as it will probably ignite many further studies in the near future. The literature is up to date and conclusions are justified by obtained results. I do not have any emerging concerns about the scientific aspects of the manuscript. However, the manuscript could be improved by making some changes as suggested as follows:

Au: Thank you very much for all your comments and suggestions. We improved the manuscript accordingly. Each specific comment is answered below.

Line 68 – Please remove table 1 with ingredients of diets into the main text.

Au: We moved the supplemental table 1 (diet composition) to the main text.

Line 98 – please change to 105oC.

Au: In the .doc file (word file) it is correct: 105oC, different from the pdf file (it is underlined). We rewrite it in the text to see if it will be corrected in the pdf file. Thank you for calling our attention.

Line 128 – please choose one version: either SHIMADZU or Shimadzu® (line 134)

Au: Thank you, we corrected it.

Line 142 – please provide a software license if necessary.

Au: We added more information about the software.

Line 176 – KEGG was already abbreviated in line 49.

Au: Thank you, we corrected it.

Line 176 – the first appearance of KO should be in line 49.

Au: Thank you, we corrected it.

Line 184 – Bacteria’s phyla should be written in italics.

Au: In the .doc file they are in italics. It was changed during pdf generation. We will double-check with the journal in the final proof of the manuscript (in case it would be accepted for publication).

Line 581 – the authors should discuss the limitations of the study at the end of the chapter.

Au: We added a paragraph in the discussion about the limitations of the study.

Line 586 – please add chapter with conclusions and future perspectives.

Au: We moved the conclusions from the end of the discussion section to a new topic “Conclusions”. We also added some future perspectives

Round 2

Reviewer 1 Report

Manuscript can be accepted provided that authors better specify whcih kind of ANOVA they used. As they still missed to specify that...."An analysis of variance (ANOVA) was performed, using the function General Linear Model, to assess the effects of supplementation and interaction with the day of measurement."

ANOVA tests are: one-way, two way and three way. Please specify along with the post hoc test used...

Author Response

Thank you for your comment. We improved the methodology by adding precision on ANOVA used: 

Ligne 215: An analysis of variance (two-way ANOVA) was performed, using the function General Linear Model, to assess the effects of supplementation and interaction with the day of measurement.